# Modulation of Bile Acid Metabolism to Improve Plasma Lipid and Lipoprotein Profiles

**DOI:** 10.3390/jcm11010004

**Published:** 2021-12-21

**Authors:** Boyan Zhang, Folkert Kuipers, Jan Freark de Boer, Jan Albert Kuivenhoven

**Affiliations:** 1Department of Pediatrics, University Medical Centre Groningen, University of Groningen, 9713 AV Groningen, The Netherlands; b.zhang@umcg.nl (B.Z.); f.kuipers@umcg.nl (F.K.); 2Department of Laboratory Medicine, University Medical Centre Groningen, University of Groningen, 9713 AV Groningen, The Netherlands

**Keywords:** bile acids, cholesterol, triglycerides, lipoproteins, bile acid sequestrants, ABST inhibitors, FXR agonists

## Abstract

New drugs targeting bile acid metabolism are currently being evaluated in clinical studies for their potential to treat cholestatic liver diseases, non-alcoholic fatty liver disease (NAFLD) and non-alcoholic steatohepatitis (NASH). Changes in bile acid metabolism, however, translate into an alteration of plasma cholesterol and triglyceride concentrations, which may also affect cardiovascular outcomes in such patients. This review attempts to gain insight into this matter and improve our understanding of the interactions between bile acid and lipid metabolism. Bile acid sequestrants (BAS), which bind bile acids in the intestine and promote their faecal excretion, have long been used in the clinic to reduce LDL cholesterol and, thereby, atherosclerotic cardiovascular disease (ASCVD) risk. However, BAS modestly but consistently increase plasma triglycerides, which is considered a causal risk factor for ASCVD. Like BAS, inhibitors of the apical sodium-dependent bile acid transporter (ASBTi’s) reduce intestinal bile acid absorption. ASBTi’s show effects that are quite similar to those obtained with BAS, which is anticipated when considering that accelerated faecal loss of bile acids is compensated by an increased hepatic synthesis of bile acids from cholesterol. Oppositely, treatment with farnesoid X receptor agonists, resulting in inhibition of bile acid synthesis, appears to be associated with increased LDL cholesterol. In conclusion, the increasing efforts to employ drugs that intervene in bile acid metabolism and signalling pathways for the treatment of metabolic diseases such as NAFLD warrants reinforcing interactions between the bile acid and lipid and lipoprotein research fields. This review may be considered as the first step in this process.

## 1. Introduction

There are numerous lines of evidence showing that changes in bile acid metabolism can affect atherosclerosis. Patients with atherosclerotic cardiovascular disease (ASCVD) have been shown to present with reduced faecal bile acid excretion [1]. In contrast, high faecal bile acid loss is associated with reduced LDL-C levels and protection against ASCVD in prospective follow-up studies [2,3]. In line, bile acid sequestrants, which increase faecal bile acid loss by inhibiting their intestinal reabsorption, leading to an increase of bile acid synthesis, were the first class of drugs approved to lower plasma cholesterol to reduce the risk of ASCVD [4,5]. Genetic evidence in humans also points to associations between bile acid and cholesterol metabolism. For example, polymorphisms in the *CYP7A1* (cytochrome P450 family 7 subfamily A member 1) gene encoding the enzyme mediating the rate-limiting step in bile acid synthesis are associated with increased levels of LDL-C and increased risk of myocardial infarction [6].

Over the last decades, it has become clear that bile acids modulate cholesterol and lipid metabolism not only as ‘catabolic products’ of cholesterol and facilitators of intestinal lipid absorption but also as signalling molecules [7]. The discovery of these hormone-like functions of bile acids has sparked the development of novel drugs that target bile acid signalling pathways [7].

In this review, we have aimed to gain insight into the effects of these (novel) drugs on plasma lipid metabolism. To introduce this topic, we first provide key information on the synthesis of bile acids and their enterohepatic circulation, their role in intestinal lipid absorption, their role as signalling molecules and their role in cholesterol catabolism, while also providing a brief description of plasma lipid and lipoprotein metabolism.

### 1.1. The Synthesis and Enterohepatic Circulation of Bile Acids

Figure 1 shows the key players in bile acid synthesis, as well as in their enterohepatic circulation. Bile acids are exclusively synthesized in the liver from cholesterol. During bile acid synthesis, hydroxyl groups are added to the cholesterol molecule that gives bile acids a hydrophilic (water-soluble) and hydrophobic (lipid-soluble) face and, hence, the amphipathic properties that make them good lipid emulsifiers. Bile acids are synthesized by two different pathways, the classical and the acidic synthesis pathways, which synthesize ~95% and ~5% of bile acids in humans, respectively (see [8] for detailed review). The first and rate-controlling step of the classical pathway is mediated by cholesterol 7α-hydroxylase (CYP7A1). A decisive step in this pathway is the hydroxylation of 7α-hydroxy-4-cholesten-3-one (C4) at the C-12 position by sterol 12α-hydroxylase (CYP8B1), which eventually determines the ratio in which the two primary bile acid species that are synthesized in humans, the trihydroxylated cholic acid (CA) and the dihydroxylated, more hydrophobic, chenodeoxycholic acid (CDCA), are produced. Bile acid synthesis via the alternative or acidic pathway generates CDCA. The acidic pathway is initiated by sterol 27-hydroxylase (CYP27A1) and involves oxysterol 7α-hydroxylase (CYP7B1).

Before being secreted into the bile by the bile salt export pump (BSEP, ABCB11; see also Figure 2), bile acids are conjugated to either glycine or taurine in an average but highly variable [9] ratio of 3:1, which increases their solubility in bile and decreases their toxicity [10]. Bile acids are then stored in the gallbladder and expelled into the intestine upon ingestion of a meal (Figure 2). In the intestine, conjugated CA and CDCA are subjected to enzymes of the gut microbiota, which catalyse deconjugation and dehydroxylation of the hydroxyl group located at the C-7 position, generating the secondary bile acid species deoxycholic acid (DCA) and lithocholic acid (LCA), respectively.

In the terminal ileum, approximately 95% of bile acids are reabsorbed. Most bile acids are taken up at the brush border membrane of enterocytes lining the lumen of the terminal ileum via the apical sodium-dependent bile salt transporter (ASBT), while small amounts are absorbed via passive diffusion from the colon. The reabsorbed bile acids are secreted into the portal blood circulation by organic solute transporter α/β (OST-α/β) and subsequently taken up by the liver via sodium taurocholate cotransporting polypeptide (NCTP) and organic anion transporting polypeptides (OATPs) for reuse [11]. This cycle is commonly referred to as the enterohepatic circulation of bile acids. Bile acids cycle between the liver and intestine ~10 times per day [12]. As ~5% of bile acids are lost at each cycle, ~40–50% of the bile acid pool needs to be replenished every day, which accounts for the conversion of ~500–1000 mg of cholesterol per day.

### 1.2. Bile Formation and Its Role in Intestinal Lipid Absorption

Bile is an iso-osmotic electrolytic fluid containing bile acids, phospholipids and cholesterol, as well as low amounts of proteins and bilirubin. Primary bile is secreted from hepatocytes into the canalicular space. From there, it will flow into the intrahepatic bile ducts, where its fluidity and alkalinity are modified by cholangiocytes, i.e., the epithelial cells lining the bile ducts [13]. Biliary bile acids are the major drivers of bile flow, inducing the secretion of water into the bile via a passive osmosis-driven transport [14]. On the other hand, bile acids are secreted into the bile by BSEP, and phospholipids and cholesterol are secreted into the bile by multidrug resistance protein 3 (MDR3) and ATP binding cassette subfamily G member 5/8 (ABCG5/G8), respectively (Figure 2) [15,16]. In the bile, bile acids and phospholipids (mainly phosphatidylcholine) form mixed micelles that also contain small amounts of cholesterol. Micelle formation reduces the detergent activity and cytotoxicity of bile acids and prevents cholesterol crystallization [17].

After the consumption of a meal, ingested lipids reach the small intestine upon their release from the stomach. The presence of dietary lipids in the duodenum triggers the release of cholecystokinin, which stimulates the contraction of the gallbladder, where bile is stored and concentrated, resulting in the release of bile into the intestinal lumen. There, bile promotes lipid emulsification, relying on the formation of mixed micelles containing bile acids, phospholipids and dietary lipids. This process is enabled by the amphipathic nature of bile acid molecules. Due to emulsification, ingested triglycerides and cholesteryl esters become more accessible for pancreatic hydrolases, which are secreted into the duodenum together with the bile from the Sphincter of Oddi. The released fatty acids and free cholesterol are subsequently transported through the unstirred water layer by the mixed micelles for uptake by the enterocytes lining the intestinal lumen. Importantly, intestinal absorption of cholesterol is fully dependent on the presence of bile acids in the intestinal lumen [18], whereas fatty acids are still absorbed in the absence of bile acids, albeit to a lesser extent [19,20].

### 1.3. Bile Acids as Signalling Molecules

In addition to their ‘classical’ role in facilitating intestinal lipid uptake, bile acids have been demonstrated to exert important signalling functions by binding to multiple nuclear and membrane-bound receptors. Over the last two decades, bile acids have been shown to activate the nuclear receptors farnesoid X receptor (FXR), pregnane X Receptor (PXR), constitutive androstane receptor (CAR) and vitamin D receptor (VDR), but also the membrane receptors Takeda G protein-coupled receptor 5 (TGR5, also known as GPBAR1), sphingosine-1-phosphate receptor 2 (S1PR2) and the muscarinic acetylcholine receptor M3 (M3R) (please see [21] for a recent review). Of these receptors, the metabolic impact of their bile acid-induced activation has been best described for FXR and TGR5, which will shortly be described below.

In 1999, FXR became the first nuclear receptor demonstrated to be activated by bile acids [22,23,24]. This receptor plays a critical role in the maintenance of bile acid homeostasis (Figure 2). In the liver, FXR prevents bile acids from reaching toxic concentrations by limiting the uptake of bile acids into hepatocytes, regulating feedback inhibition on bile acid synthesis and by stimulating the biliary secretion of bile acids via BSEP. Furthermore, in the intestine, activation of FXR induces expression of fibroblast growth factor 19 (FGF19), which is secreted into the bloodstream and then activates FGF receptor 4 (FGFR4) and its co-receptor β-klotho in the liver, leading to inhibition of bile acid synthesis via as-yet incompletely elucidated mechanisms [25]. The potency to activate FXR differs considerably between bile acid species, with CDCA > DCA = LCA > CA [22]. More hydrophilic bile acid species, such as ursodeoxycholic acid (UDCA) and mouse/rat-specific muricholic acids (MCAs) do not activate FXR. The latter has even been demonstrated to exert antagonistic activity towards FXR.

By regulating the conversion of cholesterol into bile acids, FXR considerably influences cholesterol homeostasis. Recent human genetic evidence indeed supports a role for FXR (encoded by the *NR1H4* gene) in plasma lipid metabolism: a common *NR1H4* variant (rs35724 G>C) was found to be linked with a higher serum cholesterol [26], while a rare missense *NR1H4* variant (pro.R436H) was found to be associated with lower cholesterol levels and protection against coronary artery disease [27]. Transcriptional profiling indicated that the R436H mutation is not a loss-of-function variant. The functionality of these *NR1H4* gene variants, however, remains to be studied further. Finally, earlier studies in mice have shown that whole-body ablation of FXR results in increased non-HDL-C, triglycerides and HDL-C [28,29].

TGR5 is highly expressed in the intestine and gallbladder but also in brown adipose tissue (BAT) and muscle [30]. In intestinal L-cells, activation of TGR5 triggers the release of glucagon-like peptide 1, which, in turn, promotes glucose-induced insulin release from the pancreas and stimulates satiety. In BAT, activation of TGR5 causes an increase in cyclic AMP, leading to increased expression of type 2 iodothyronine deiodinase (DIO2), which mediates the conversion of the inactive thyroid hormone T4 into active T3, resulting in increased energy expenditure [31]. BAT activity is associated with active uptake of free fatty acids, as well as lipoprotein remnants, and hence represents a determinant of plasma lipid levels. In mice, pharmacological activation of TGR5 reduces hepatic lipid content and plasma triglyceride levels [32]. Moreover, bile acid sequestration has recently been demonstrated to enhance the beneficial effects of BAT activation on hyperlipidaemia and atherosclerosis development in mice with humanized lipoprotein metabolism, i.e., APOE*3-Leiden.CETP-transgenic mice [33]. The order of potency of bile acids to activate TGR5 is LCA > DCA > CDCA > CA [34].

### 1.4. Cholesterol, Triglycerides and Plasma Lipoprotein Metabolism

The cholesterol molecule is the building block of bile acids and steroid hormones but also plays a vital role in membrane structure and fluidity. While cholesterol can be obtained from dietary sources, approximately 80% of the cholesterol present in the human body is de novo synthesized [35,36]. Almost all cells can synthesize cholesterol, which emphasizes its key role in cellular homeostasis. It is important to note that cholesterol cannot be degraded in mammalian systems and that its turnover relies on the sloughing of (dead) cells from the skin or the intestinal wall and on the conversion into steroids that are metabolized and eventually removed from the body. Relatively small amounts of cholesterol are used for the synthesis of steroid hormones. The only quantitatively meaningful pathway of cholesterol catabolism is through the synthesis of bile acids in the liver. The adult human body contains about 2 g of bile acids [12]. Each day, about 0.5–1.0 g of bile acids leave the body with faeces. This amount is replenished by de novo synthesis from the cholesterol [37,38].

Similar to cholesterol, triglycerides are also taken up from the diet in the gut but only after their hydrolysis into glycerol and free fatty acids, a process that is facilitated by bile acids as described below (Figure 3). In the postprandial phase, resynthesized triglycerides in intestinal cells are packed into chylomicrons. These largest of apolipoprotein B (apoB)-containing lipoproteins are rich in triglycerides and low in cholesterol. Chylomicrons are secreted into the lymph, which reaches systemic blood circulation via the thoracic duct. In the fed state, the rapid lipolysis of triglycerides in chylomicrons in the periphery ensures delivery of free fatty acids for energy production in parenchymal tissue in, for example, the heart and skeletal muscle and for energy storage in adipose tissue. Following triglyceride hydrolysis, chylomicron remnants—which are rich in cholesterol—are taken up by the liver.

Next to the above-described exogenous pathway, the endogenous lipoprotein pathway starts with the production of triglyceride-rich very-low-density lipoproteins (VLDL) in the liver. In the fasted state, the liver ensures sufficient energy supply from various sources to the periphery. Dependent on the metabolic state, it can, for example, obtain fatty acids from the periphery, release triglycerides from intracellular lipid stores or ensure triglyceride supply through de novo lipogenesis for VLDL synthesis and secretion into the blood circulation. As in chylomicrons, the triglycerides in VLDL are hydrolyzed in the periphery, rendering smaller-sized lipoproteins that eventually become cholesterol-rich LDL, which are primarily taken up by the liver.

The small intestine and liver finally also secrete the much smaller high-density lipoproteins (HDL), which carry apoA-I as major apolipoprotein. HDL only carry very small amounts of triglycerides compared to chylomicrons and VLDL and are instead rich in cholesteryl esters. HDL does not serve major metabolic functions but HDL-C concentration is inversely related to plasma triglyceride levels and risk of ASCVD [39]. However, the association of HDL-C with the risk of ASCVD is not causal, as evidenced by Mendelian randomization studies, as well as the failure of drugs that have been used to target HDL metabolism to decrease the risk of ASCVD [40,41,42].

Due to their bactericidal properties, bile acids fulfill an important role in shaping the gut microbiome [43]. Vice versa, gut bacteria impact the composition of the bile acid pool by deconjugating and dehydroxylating bile acids. A recent study shows that gut bacteria, known to be involved in bile acid metabolism, are associated with the interindividual variation of BMI, plasma triglycerides and HDL-C [44]. Although indirect, these data illustrate yet another link between bile acid and lipid metabolism.

### 1.5. Challenges and Aim of This Review

The molecular understanding of bile acid metabolism and signaling is largely obtained through studies in mice. This poses major challenges when it comes to understanding the mechanisms that may explain the effects of bile acid modulators on human plasma lipid and lipoprotein metabolism. Foremost, murine and human bile acid metabolism are very different because of the unique presence of muricholic acids in mice. These extremely hydrophilic trihydroxylated bile acid species are efficiently synthesized from CDCA in mice by the enzyme CYP2C70 [45], mainly by sequential 6β-hydroxylation, generating αMCA and epimerization of the hydroxyl group at C-7 from the 7α to the 7β orientation, generating βMCA [46] as primary bile acids in these animals. Bacteria in the gut then generate the secondary ωMCA. MCAs constitute about 30–40% of murine bile acids [11] and have a considerable impact on the physicochemical characteristics of the murine bile acid pool. Because of their extremely hydrophilic nature, MCAs have inferior lipid solubilizing capacity compared to the bile acid species in humans [47], which hampers the translation of preclinical data, especially when interventions impact bile acid pool composition as is, for example, seen upon FXR stimulation [46].

Extrapolation of murine data is further complicated by the fact that CDCA is the most potent endogenous FXR agonist in humans, while taurine-conjugated α/βMCA in mice, by contrast, exert antagonistic FXR activity [48]. Several groups have recently generated *Cyp2c70*-deficient mice with a human-like bile acid composition [46,49,50,51] to improve the translation of preclinical data.

On top of the differences in bile acid metabolism between mice and humans, the lack of cholesterol ester transfer protein (CETP) in mice poses a second major challenge because, as a result, mice carry most of their plasma cholesterol in HDL. Humans instead carry most of their cholesterol in LDL as CETP drives the transfer of cholesteryl esters from HDL to apoB-containing lipoproteins in exchange for triglycerides. While the CETP reaction in humans is driven by the plasma triglyceride concentration [52], mice are compared to humans with very fast VLDL metabolizers, which further complicates translation. Taking all the above into consideration, we have, for this review, decided to keep speculations on the possible mechanisms that may underlie the associations observed in clinical trials to a minimum.

In this review, we have focused on trials in which bile acid modulating drugs have been used in humans in a (first) step to bridge the gap between the bile acid and plasma lipid metabolism research fields by characterizing the effects of these drugs on plasma lipid metabolism.

## 2. Intervening in the Enterohepatic Circulation of Bile Acids

### 2.1. Bile Acid Sequestrants

#### 2.1.1. Mechanism of Action

Faecal bile acid loss can be increased pharmacologically by bile acid sequestrants (BAS). These are basic anion-exchange resins, non-absorbable polymeric molecules that bind negatively charged bile acids in the intestine. By interfering with bile acid re-absorption, BAS divert bile acids from the enterohepatic cycle and, hence, promote their faecal loss. In order to maintain a stable bile acid pool size, a synthesis of bile acids from cholesterol is induced, which is reflected by elevated levels of the bile acid synthesis intermediate C4 in the plasma [53,54,55]. This, in turn, leads to a reduction of the cholesterol content in membranes of the endoplasmic reticulum (ER) of hepatocytes and, hence, to a conformational change of the sterol regulatory element-binding protein cleavage-activating protein (SCAP) that causes its release from the ER-anchored insulin-induced gene (INSIG)-1 or -2 [56]. SCAP then escorts sterol regulatory element-binding protein 2 (SREBP2) during transport from the ER to the Golgi, where it undergoes activating proteolytic cleavage. This cleaved SREBP2 then travels to the nucleus to induce expression of its target genes such as *HMGCR*, encoding 3-hydroxy-3-methyl-glutaryl-coenzyme A reductase, the target of statins, to increase cholesterol synthesis, and *LDLR* (encoding the LDL receptor), to increase hepatic cholesterol import. Thus, interruption of enterohepatic recirculation of bile acids translates into effects on hepatic cholesterol and lipoprotein metabolism, including stimulation of LDLR-mediated hepatic LDL-C uptake from the blood circulation [57,58]. Notably, cholestyramine was the first drug to demonstrate LDL-C lowering and protection against coronary heart disease in a randomized clinical trial that included 3806 asymptomatic middle-aged men with primary hypercholesterolemia [4,5].

#### 2.1.2. BAS as Monotherapy

BAS have been used as monotherapy in patients with dyslipidemia, type 2 diabetes (TD2), or both. Four main bile acid sequestrants are currently available for use in the clinic: cholestyramine (Questran, 1973), colestipol (Colestid, 1977), colestilan (BindRen, 1999) and colesevelam (Welchol, 2000). Cholestyramine and colestipol are hard to ingest and give unwanted side effects, such as bloating [59]. The second-generation BAS colestilan and colesevelam are better tolerated. We have summarized the results of the main recent human studies in Table 1 to get a better grasp on the effects of BAS treatment on plasma lipids in different patient groups.


*Treating familial hypercholesterolemia (FH)*


In both adult and pediatric patients with heterozygous FH, colesevelam is efficacious and well-tolerated [60,61]. To illustrate the effects of BAS on dyslipidemia as monotherapy: in a double-blind study in 2010, 194 children with FH were randomized to colesevelam or placebo. The use of colesevelam for 8 weeks was associated with significant reductions of total cholesterol (−7.4%), LDL-C (−12.5%) and apoB (−8.3%), but increased HDL-C (+6.1%) and triglycerides (+5.1%), compared to the placebo [61].

At very high dosages, cholestyramine has also been studied as monotherapy in FH patients (*n* = 26) which was shown to affect subclinical atherosclerosis. Combined with a fat-modified diet, 8 g of cholestyramine twice daily has recently (2016) also been shown to improve the course of coronary atherosclerosis, as assessed with surrogate clinical endpoints. These changes were seen in the context of marked reductions in total cholesterol (−26.2%) and LDL-C (−35.4%), compared to care with only a fat-modified diet [62]. In the treatment arm, non-significant changes in triglycerides (+8.0%) and HDL-C (−3.1%) were noted.


*Treating type 2 diabetes (T2D)*


Since 2008, BAS have also been used in patients with T2D because of their favorable effects on glucose metabolism [57]. Here, we have summarized several recent studies showing the effect of BAS as monotherapy in a T2D setting.

In 2010, the use of 4.5 g/day colestilan, a BAS which binds both phosphate and bile acids, in 183 patients with T2D for 12 weeks, improved glycemic control and significantly reduced LDL-C levels (−22.5%) and increased HDL-C (+6.6%) without changing triglycerides [63]. In 2012, a randomized, double-blind, placebo-controlled study showed the effects of colesevelam (3.75 g/d) in 216 patients with prediabetes and primary hyperlipidemia (LDL-C ≥ 100 mg/dL and triglycerides < 500 mg/dL). At the end of the 16-week study, total cholesterol, LDL-C and apoB were significantly reduced by 7.2%, 15.6% and 8.1%, respectively, with colesevelam versus placebo. However, the treatment arm also presented with a significant increase in triglycerides (+14.3%) [64]. In 2013, the use of colesevelam (3.75 g/day) in 176 T2D patients for 24 weeks was shown to reduce total cholesterol, LDL-C and apoB by −5.1%, −11.2% and −6.5%, respectively, compared to 181 patients treated with placebo [65]. Also in this study, triglycerides were increased (+9.7%) in the colesevelam-treated group.

To summarize, the use of BAS in FH and T2D patients results in moderate lowering of LDL-C and apoB, which generally coincides with a modest increase in plasma triglyceride levels. The latter finding may be explained by a combination of factors, including interference with FXR signalling and induction of bile acid synthesis. It has, in this regard, been shown that familial hypertriglyceridemia is frequently associated with a high hepatic bile acid synthesis rate [66,67]. Accordingly, induction of bile acid synthesis by bile acid sequestration positively correlates with increases in plasma triglyceride levels [60]. In mice, it has been shown that FXR-deficiency is associated with elevated plasma triglyceride levels [68,69], while FXR activation has been demonstrated to reduce plasma triglycerides [70,71,72]. BAS treatment reduces FXR activation, resulting in tapered FXR-mediated suppression of the expression of the lipogenic transcription factor SREBF1C [73,74]. However, actual inhibition of lipogenesis was not detected in a clinical study with colesevelam [75]. An alternative explanation is that *APOC2* is a target of FXR. This gene encodes for apoC-II, which is a cofactor of lipoprotein lipase, the enzyme that can lipolyses triglycerides in lipoproteins. Reduced activation of FXR upon BAS treatment may decrease VLDL-associated apoC-II, leading to reduced lipolysis of VLDL-triglycerides and, hence, increased plasma triglyceride concentrations. Additionally, phosphatidic acid phosphatase is activated under abnormal bile acid enterohepatic circulation caused by BAS and may increase triglyceride synthesis by converting phosphatidic acid into a diglyceride [76,77]. However, the mechanisms that cause plasma triglycerides to increase in humans upon the use of BAS remain to be established.

#### 2.1.3. Bile Acid Sequestrants Combined with Other Drugs


*Treating primary dyslipidemia*


Statins are HMG-CoA reductase inhibitors that reduce cholesterol biosynthesis by blocking the mevalonate pathway. They are the first-line therapy for reducing LDL-C, but there are patients that are either intolerant to statin therapy or do not reach their LDL-C targets according to the guidelines for the ASCVD risk treatment [78]. In this context, statins are regularly combined with BAS in patients with dyslipidemia. The studies listed below give an impression of the sizes of the treatment effects.

In patients with moderate hypercholesterolemia (LDL-C, 160–220 mg/dL, and triglycerides ≤ 300 mg/dL), the group treated with a combination of colesevelam (2.3 g/d) and lovastatin (10 mg/d) for 4 weeks showed decreased total cholesterol (−21%), LDL-C (−34%) and apoB (−24%). The effects of the combined treatment were superior to those observed when either agent was used alone (lovastatin decreased total cholesterol and LDL-C by 15% and 22%, respectively) [79]. These effects were seen without significant changes in HDL-C and triglycerides. In another study, 258 patients with primary hypercholesterolemia (LDL-C ≥ 160 mg/dL and triglycerides ≤ 300 mg/dL) were treated with colesevelam and simvastatin, or colesevelam or simvastatin alone for 6 weeks. Subjects treated with combination therapy (colesevelam 3.8 g with simvastatin 10 mg, or colesevelam 2.3 g with simvastatin 20 mg) showed a mean reduction of LDL-C by 42%, which exceeded the reductions on simvastatin 10 mg (−26%) or 20 mg (−34%), or, for colesevelam, 2.3 g (−8%) or 3.8 g (−16%) alone [80]. Again, HDL-C and triglyceride levels were not different between groups.

In a recent meta-analysis (2020) comparing statin monotherapy with a combination of statin and BAS (a total of 1324 patients with dyslipidemia), it was demonstrated that BAS lead to an additional LDL-C reduction of 16.2% when used on top of statins [81] but no data on other plasma lipids were reported.


*Treating type 2 diabetes*


The dual favorable effects of colesevelam on both glucose metabolism and lipid metabolism are underlined by direct comparisons with other antidiabetic agents. For 169 patients with T2D insufficiently, controlled with metformin monotherapy (≥3 months), colesevelam (*n* = 57), the peroxisome proliferator-activated receptor γ (PPARγ) agonist rosiglitazone maleate (*n* = 56) and the dipeptidyl peptidase-4 (DPP-4) inhibitor sitagliptin phosphate (*n* = 56) all significantly improved glycemic control, whereas only colesevelam significantly reduced levels of LDL-C (−11.6%) with slight reductions of total cholesterol (−2.9%) and non-HDL-C (−3.8%) but significantly increased triglycerides (+14.9%) [82]. The use of the PPARγ agonist pioglitazone has also been combined with colesevelam (2014): 562 T2D patients with suboptimal glycemic control were randomized to colesevelam 3.8 g/day or placebo added to existing stable pioglitazone-based therapy [83]. Compared with placebo, colesevelam added to pioglitazone improved glycemic control, decreased total cholesterol (−6.5%), LDL-C (−16.4%), non-HDL-C (−9.8%) and apoB (−8.8%), but increased triglycerides (+11.3%) after 24 weeks of treatment.

A post-hoc analysis on 696 patients with T2D from three randomized, double-blind, placebo-controlled studies showed that adding colesevelam (3.75 g/d) to metformin compared to placebo significantly decreased HbA1c (−0.5%) and fasting plasma glucose (−15.7 mg/dL), and reduced levels of total cholesterol (−5.8%), LDL-C (−16.5%), apoB (−7.6%) and non-HDL-C (−8.2%), but triglyceride levels increased (+12.8%) [84]. Another study that compared metformin plus placebo to metformin plus colesevelam (3.75 g/d) in 286 patients with T2D showed almost identical results: significantly decreased HbA1c (−0.3%), total cholesterol (−6.1%), LDL-C (−16.3%), non-HDL-C (−8.3%) and apoB (−8.0%), while triglycerides increased (+18.6%) [85].

In a meta-analysis, it was finally shown that BAS treatment in patients with T2D reduces LDL cholesterol, although to a lesser extent than statin treatment. While three of the studies (553 patients) included in the analysis showed non-significant increases in triglycerides, five trials (1709 patients) showed significantly increased plasma triglyceride levels upon colesevelam treatment (14.1–21.5%) [86].


*Combinations of BAS with ezetimibe and statins*


Besides statins, ezetimibe, which inhibits the absorption of cholesterol at the small intestinal brush border via inhibition of the sterol transporter Niemann-Pick C-1-Like 1 (NPC1L1) [87], has been shown to attenuate the increase in plasma triglycerides induced by BAS: In a small but interesting study with twelve patients with T2D and four patients with metabolic syndrome with a history of statin intolerance, colesevelam, combined with ezetimibe (3 months), markedly affected triglycerides. The combination therapy was associated with significant reductions of total cholesterol (−27.5%), LDL-C (−42.2%) and non-HDL-C (−37.1%), as well as a decrease in triglycerides (−30.8%), compared to the baseline [88]. Ezetimibe has, by itself, also been demonstrated to have a favorable impact on triglyceride metabolism in T2D patients who are also treated with simvastatin [89] and in obese hypercholesterolemic patients [90].

On the other hand, colesevelam combined with ezetimibe and a statin can render a net-zero effect on plasma triglycerides: 86 patients with FH were randomly assigned to receive colesevelam (3.75 g/d) or a placebo added to the statin plus ezetimibe for 12 weeks. Colesevelam added to a statin and ezetimibe provided an additional reduction of total cholesterol (−7.3%) and LDL-C (−12.0%), while HDL-C and triglycerides were not significantly affected [91].

Finally, high dosages of BAS do not further decrease cholesterol when added to a high-dose statin treatment: in 144 patients with severe hypercholesterolemia (LDL-C, 190–400 mg/dL), the addition of cholestyramine (16 g/d) to rosuvastatin (80 mg/d) did not translate into additional lowering of LDL-C compared to rosuvastatin alone [92].

Taken together, BAS moderately but rather consistently increase plasma triglycerides when applied as monotherapy in both FH and T2D patients. It has been demonstrated in obese subjects with and without T2D that BAS monotherapy does not alter the size of the circulating BA pool. Instead, colesevelam induces a shift in its composition towards trihydroxylated CA at the expense of dihydroxylated DCA and CDCA [74]. Interestingly, the increase in plasma triglycerides was linearly related to the hepatic bile acid synthesis rate, underscoring the relevance of microsomal cholesterol depletion in control of plasma triglyceride levels in humans [74]. However, when BAS are combined with statins or ezetimibe, the increase of plasma triglycerides is attenuated [79,80,88,91]. This may be due to the notion that statins, as well as ezetimibe, can slightly reduce triglyceride levels [89,93,94]. On the other hand, when BAS are combined with glucose-sensitizing drugs in T2D patients, the effects on triglyceride levels are unaltered [82,83,84,85].

### 2.2. ASBT Inhibitors

As indicated in the introduction and Figure 2, ASBT (the apical sodium-dependent bile acid transporter, ileal bile acid transporter or ileal sodium-dependent bile acid transporter) actively absorbs bile acids from the gut lumen [95] and hence fulfills a crucial step in the enterohepatic circulation. This makes ASBT an attractive target for intervention [96]. ASBT inhibitors have so far primarily been studied for the treatment of bile acid-related pathologies such as chronic constipation and pruritus in primary biliary cholangitis (PBC) and genetic cholestatic liver diseases [97], but we have screened the literature specifically for data that may help to shed light on how ASBT inhibitors impact plasma lipid levels.

#### 2.2.1. Elobixibat (A3309)

In 2011, elobixibat was used in patients with chronic idiopathic constipation to evaluate effects in two double-blind, placebo-controlled studies [98,99]. In the first study, performed in 30 patients, elobixibat (10 mg/d) treatment for 14 days markedly increased C4, a proxy for bile acid synthesis, which was associated with reductions in total cholesterol (−11%) and LDL-C (−17%) [98]. In the second study, 36 patients with chronic constipation with normal lipid levels at baseline were treated with elobixibat (15 or 20 mg/d). After 14 days of use, elobixibat increased C4 levels but did not significantly affect total cholesterol and LDL-C [99]. In 2018, two much larger trials were conducted in patients with chronic constipation [100]. Effects on plasma LDL-C and HDL-C were only reported for the short term (14 days) trial in this study. LDL-C was found to be significantly reduced upon elobixibat (10 mg/d) treatment, whereas HDL-C remained unchanged [100] and no data on triglycerides were reported. Another study conducted in 2018 involved 60 patients with chronic constipation who were randomized to treatment with five dosages of elobixibat (2.5, 5, 10, 15 or 20 mg/d, in groups of 10 patients). Elobixibat was again correlated with higher C4 levels and reduced LDL-C, while HDL-C was non-significantly reduced (no data on triglycerides provided) [101].

#### 2.2.2. Linerixibat and Odevixibat

The ASBT inhibitor linerixibat (GSK2330672) has been tested in 75 patients with T2D to investigate the effects on glucose and plasma lipids. Compared with placebo, the 14-day use of 90 mg linerixibat twice daily reduced fasting plasma glucose (−1.21 mmol/L), fasting total cholesterol (−23.8%), LDL-C (−31.3%), non-HDL-C (−28.3%) and apoB (−26.3%), without changing HDL-C. A trend towards increased triglyceride levels was observed in the linerixibat group [102]. In 2016, odevixibat (A4250), another novel ASBT inhibitor, was tested in 40 healthy individuals. This resulted in a significant increase in faecal bile acids and plasma C4 but, unfortunately, plasma lipids were not measured [103]. In a small subsequent study, the same compound was used in nine PBC patients, all of whom reported a remarkable improvement in pruritus, but no changes in plasma lipid levels were observed [104].

Taken together, the use of ASBT inhibitors appears to have favorable results on plasma lipids. However, the studies are small, may have been impacted by the underlying pathology of the selected patients and data on lipids are not always reported, making it difficult to draw firm conclusions regarding the impact of ASBT inhibition on plasma lipids. Nevertheless, most data support the idea that ASBT inhibitors and BAS have similar effects on lipid metabolism. Getting improved insight into this matter may be relevant when considering that, for example, chronic obstipation is a marker for increased cardiovascular risk in post-menopausal women [105].

### 2.3. Bile-Salt Export Pump (BSEP), Organic Solute Transporter-α/β (OST-α/β) and Sodium Dependent Taurocholate Cotransport Peptide (NTCP)

Next to ASBT, three other main bile acid transporters, i.e., BSEP, OST-α/β and NTCP, play key roles in the enterohepatic circulation (Figure 2), and we have screened the literature for their possible impact on plasma lipid traits.

*BSEP.* BSEP is an ATP-dependent membrane transport protein functioning in the canalicular membrane of hepatocytes by actively secreting bile acids into the canalicular space [106]. Individuals with BSEP deficiency display progressive familial intrahepatic cholestasis type 2 (PFIC2), a severe cholestatic condition often necessitating liver transplantation at paediatric age for survival. Next to PFIC, a less severe form of cholestasis occurs in patients with BSEP mutations that are associated with substantial residual bile acid transport activity. This condition is called benign recurrent intrahepatic cholestasis type 2 (BRIC2) [107,108]. It is, however, not reported whether severe or less severe mutations in BSEP are associated with changes in plasma lipid levels. The generally severe clinical phenotype of these patients is likely leaving little room for concerns of dyslipidemia. Activation of BSEP has been considered an option for treating acquired liver diseases such as drug-induced liver injury and intrahepatic cholestasis [106] but such studies have, according to our knowledge, not been conducted thus far.

Interestingly, BSEP knockout mice do not display overt cholestasis as they, in sharp contrast to BSEP-deficient humans, are able to maintain a substantial biliary bile acid secretion [109], which underlines that findings in these mice are difficult to translate to humans. Hepatic overexpression of *Abcb11* encoding for BSEP in mice has been shown to promote high-fat diet-induced obesity, as well as high-cholesterol diet-induced hypercholesterolemia [110], and it may thus be interesting to study the relation between BSEP and plasma lipids in more depth.

*OST-α/β.* The subunits of OST-α/β (encoded by the *SLC51A* and *SLC51B* genes) form a heteromeric solute carrier protein that transports bile acids, steroid metabolites and drugs out of cells. OST-α/β protein expression is highest in the intestinal tract and facilitates bile acid transport from the gut to the portal system [111], but it is also expressed in the liver. OST-α/β protein expression in the liver is significantly increased in patients with extrahepatic cholestasis [112], PBC [113] or NASH [114,115]. Only recently, the first case of OST-α deficiency was identified [116]: a boy carrying homozygous for an *SLC51B* variant (c.79delT, p.F27fs) suffered from cholestasis, liver fibrosis and congenital diarrhea, but there is a lack of data on plasma lipids in this study. In 2018, two brothers with OST-β deficiency (homozygosity for a single nucleotide deletion in codon 27 of *SLC51B*), who suffered from congenital diarrhea and cholestasis [117], were identified. However, plasma levels of cholesterol and triglycerides were in normal ranges.

*NTCP.* The multi-transmembrane glycoprotein NTCP, encoded by the *SLC10A1* gene, is predominantly expressed on hepatocytic basolateral membranes and transports particularly conjugated bile acids from the portal system into hepatocytes. NTCP is expressed throughout the liver acinus and at the luminal membrane of pancreatic acinar cells [118]. The first NTCP-deficient mouse model was established in 2015 and showed markedly decreased clearance of serum bile acid concentrations [119]. Interestingly, only a subset of NTCP-deficient mice became hypercholanemic and only those mice showed reduced faecal bile acid excretion, while the normocholanemic NTCP-deficient mice had similar faecal bile acid excretion as the wild-type controls. Plasma lipid levels were, however, not reported in this study. In 2015, the first patient with genetic NTCP deficiency was identified, shown to suffer from mild hypotonia, growth retardation, delayed motor milestones and conjugated hypercholanemia, i.e., elevated plasma bile acid concentrations. Plasma lipid levels changed markedly in this patient at 2.9–5.3 years of age [120]. Since then, several other cases of hypercholanemia due to mutations in NTCP have been reported but without data on plasma lipid traits [121,122].

NTCP inhibitors have been developed for inhibiting the hepatic entry of hepatitis B and D [123]. A mouse study showed that the NTCP inhibitor myrcludex B significantly reduced plasma cholesterol and increased triglycerides in high-fat diet-fed *Oatp1a/1b* KO mice [124]. In humans, myrcludex B has been shown to increase total plasma bile acids about 18-fold without inducing signs of cholestasis in healthy volunteers [125]. Effects on plasma lipid metabolism were, however, not reported in this study.

Taken together, the amount of information available in the literature concerning the impact of these key regulators of bile acid (re)circulation on plasma lipid metabolism is, as yet, very limited.

## 3. Bile Acid Synthesis and Plasma Lipids

In this section, we describe the effects of drugs targeting bile acid-signaling pathways and their effects on plasma lipids.

### 3.1. FXR and TGR5 Agonists

#### 3.1.1. FXR Agonists

*Obeticholic acid (OCA, OCALIVA).* OCA is a synthetic derivative of CDCA with potent FXR agonistic activity. This compound has been approved by the FDA for the treatment of PBC in combination with ursodeoxycholic acid (UDCA) in adult patients with an inadequate response to UDCA. As FXR agonists reduce hepatic bile acid synthesis, its effects on plasma lipids could be anticipated to be opposite to those seen with BAS, i.e., an increase of LDL-C, due to reduced usage of cholesterol for bile acid production, and a decrease in triglycerides. However, FXR also regulates the expression of lipid-modulating genes, which also contribute to the overall effects.

OCA has also been tested in NAFLD/NASH, and NASH-related fibrosis [126]. In 2013, a double-blind, placebo-controlled study evaluated the effects of OCA on insulin sensitivity in 64 patients with T2D and NAFLD. Treatment with 50 mg daily reduced C4 levels, increased insulin sensitivity and improved liver function compared to the placebo. These favorable effects were accompanied by increased total cholesterol (+7%) and LDL-C (+24%), decreased HDL-C (−14%) and decreased triglycerides (−23%) [127].

In 2015, a phase 2b randomized, placebo-controlled trial with OCA was conducted with 283 patients suffering from non-cirrhotic NASH with normal baseline levels of total cholesterol, LDL-C and HDL-C. After daily treatment with 25 mg OCA for 72 weeks, the NAFLD activity score improved, compared to the placebo. Compared to the placebo, the levels of total cholesterol and LDL-C increased by 7.9% and 15.5%, respectively, while HDL-C decreased by 5.5% and triglyceride levels were not changed in the OCA group [128]. In a subsequent phase 3 study in patients with NASH (*n* = 1968), OCA (10 mg or 25 mg daily) also increased LDL-C, especially during the early phase of the study [126]. Plasma LDL-C gradually decreased after the initial increase but was still somewhat elevated by month 18. HDL-C decreased in a dose-dependent manner and remained consistently reduced throughout the study, while triglycerides were reduced mainly in the group receiving the 25 mg dose (−21% at month 18).

OCA has also been administered to 28 patients with bile acid diarrhea but normal cholesterol and triglycerides. After receiving oral OCA 25 mg daily for two weeks, symptoms of abdominal pain, urgency and bloating were improved, while plasma FGF19 was increased and bile acid synthesis was reduced. Again, total cholesterol and LDL-C increased by 10.4% and 19.7%, respectively, whereas there were no significant effects on the triglycerides [129]. Another study, performed on 68 healthy individuals, showed that OCA increased LDL-C and decreased HDL-C [130]. In the CONTROL trial, the OCA treatment was combined with atorvastatin in 67 patients with NASH to mitigate the effects of FXR activation on plasma LDL-C concentrations. Four weeks of treatment with OCA (5, 10 or 25 mg) alone caused significant increases in plasma LDL-C, which rapidly declined below baseline levels when atorvastatin was added to the treatment regimen [131].

Combined, these findings indicate that FXR activation by OCA has opposite effects compared to the use of BAS regarding LDL-C. The same appears to be true for changes in triglycerides and HDL-C, i.e., increases with BAS and decreases with OCA, but the results are not completely consistent.

*Non-steroidal FXR agonists.* PX-102, a non-steroidal FXR agonist, induced a >80% decrease in C4 after an 8 h single administration in healthy male volunteers [132]. However, serum cholesterol remained unchanged up to 24 h after administration, while triglycerides tended to decline in this study. Clinical development of PX-102 itself has been stopped, but the next generation, structurally related, compound cilofexor (GS-9674, PX-201) is being tested for its efficacy in NASH. The efficacy and safety have been evaluated in a phase 2 trial that included 140 patients with non-cirrhotic NASH [133], in which treatment with 30 or 100 mg/d cilofexor for 24 weeks did not impact plasma lipids. Cilofexor has also been used in patients with bridging fibrosis or compensated cirrhosis (F3-F4) attributable to NASH. Low dose Cilofexor (30 mg/d) monotherapy did not induce significant changes in plasma triglycerides and LDL-C following 48 weeks of treatment [134].

The structural ancestor of PX-102 and cilofexor, GW4064, has been shown to reduce postprandial plasma cholesterol and triglyceride levels in mice [135]. However, GW4064 has never been used in a clinical setting because of poor bioavailability and potential hepatotoxicity [136]. The oral bioavailability of GW4064 could be substantially improved using a self-emulsifying drug delivery system [137]. Using this approach, it was found that GW4064 decreased LDL-C in high-fat/high-cholesterol-fed hamsters, while HDL-C and triglycerides were not significantly affected. In the same study, treatment of normal diet-fed, as well as high-fat/high-cholesterol-fed, cynomolgus monkeys with GW4064 for 4 weeks resulted in decreased HDL-C and increased triglycerides, while non-HDL-C levels were not significantly changed. The non-steroidal FXR agonist tropifexor (LJN452) was shown to not significantly affect plasma triglycerides, total cholesterol, HDL-C and LDL-C in a small clinical study assessing its safety, tolerability, pharmacokinetics and pharmacodynamics [138].

In a randomized, double-blind, placebo-controlled phase I study, TERN-101 (another non-steroidal FXR agonist formerly known as LY2562175) dose-dependently decreased plasma C4 levels up to 91% during a 7-day treatment period, without having appreciable effects on plasma LDL-C levels [139]. MET-409, a fexaramine-derived FXR agonist decreased liver fat content at dosages of 50 and 80 mg/d [140]. LDL-C levels were significantly elevated (+24%) in the 80 mg/d group only, while an insignificant 6.8% increase was observed in the group receiving 50 mg/d. HDL-C was, however, decreased with both dosages, −20% and −23% for the 50 and 80 mg/d doses, respectively [140]. Data on triglycerides were not reported in this study. Finally, a novel non-bile acid class FXR agonist containing steroid and non-steroid components was recently shown to reduce liver fat in non-cirrhotic patients with fibrotic NASH at 2.5 mg/d. Patients receiving 2.5 mg/d EDP-305 (*n* = 53) showed decreased HDL-C, increased apoB concentrations and a strong trend towards increased LDL-C compared to placebo (*n* = 24), while plasma triglyceride levels were not affected [141].

Taken together, the FXR studies addressed in this review show diverse effects on plasma lipids. OCA is the best-studied compound thus far and has been shown to have beneficial effects on NAFLD/NASH, but it induces a potentially atherogenic lipoprotein profile characterized by increased LDL-C and decreased HDL-C. Plasma triglycerides may decrease upon treatment, but this effect is not consistently observed. Non-steroidal FXR agonists appear to cause fewer adverse effects on plasma lipid levels in humans. However, these peculiar observations may be related to the relatively early phase of clinical studies with these compounds, generally including limited numbers of participants. Larger longer-term studies will be required to more thoroughly assess the effects of (non-steroidal) FXR agonists on plasma lipids profiles. The dosing regimens will likely need to be carefully optimized to improve liver function in NAFLD/NASH patients while minimizing the unwanted changes in the plasma lipids [142]. Combination treatment with lipid-lowering drugs may be necessary. Moreover, the data available at this moment suggests that structural optimization of FXR agonists could potentially reduce the adverse effects of FXR activation on plasma lipids.

#### 3.1.2. TGR5 Agonists

As described in the introduction, the G protein-coupled bile acid receptor 1 (*GPBAR1*) gene, encoding for TGR5, is expressed in metabolically active tissues such as the liver, intestine and gallbladder, as well as in immune cells. Bile acids act as natural ligands for this GPCR. Interestingly, TGR5 activation with the synthetic bile acid analogue INT-777 was shown to protect *Ldlr*^−/−^ mice against atherosclerosis [143] by reducing macrophage inflammation and lipid loading. Based on several murine studies [144,145,146], TGR5 agonists are being developed to treat T2D and steatohepatitis [147]. In humans, data on TGR5 are thus far mostly limited to genome-wide association studies, which link the gene locus with ulcerative colitis and primary sclerosing cholangitis [148,149]. In 2010, sequencing of *GPBAR1* in 267 patients with primary sclerosing cholangitis (PSC) rendered five non-synonymous mutations that can reduce TGR5 function, but no data on plasma lipid parameters were reported [150]. So far, only one TGR5 agonist, SB-756050, has been used in patients with T2D and shown to cause an unexpected increase in glucose excursions in patients after an oral glucose challenge but data on lipids were not reported [151].

## 4. Conclusions and Perspectives

In this review, we set out to improve our understanding of the relationship between bile acid metabolism and plasma lipid homeostasis by providing a comprehensive overview of the effects of drugs that modulate bile acid metabolism and signaling pathways. We have mainly focused on the effects of pharmaceutical interventions in humans for the reasons outlined in the introduction (see challenges).

In Table 1 and Table 2, we have summarized our main findings. Data from available literature uniformly indicate that increasing bile acid synthesis with BAS reduces plasma total cholesterol and LDL-C levels, with variable effects on HDL-C. However, moderate but fairly consistent increases in plasma triglyceride levels are observed upon BAS treatment. The effects of ASBT inhibitors are quite similar to those obtained with BAS, which could be expected because both drug classes increase faecal bile acid loss. However, the ASBT inhibition studies carried out so far only include small patient cohorts with various underlying pathologies, which hampers drawing firm conclusions at this moment. Of the FXR agonists that have been clinically evaluated, OCA is studied most extensively. This drug reduces bile acid synthesis and is therefore expected to render opposite results compared to BAS. Indeed, OCA increases plasma total cholesterol levels and, in particular, LDL-C. Combination treatment with statins has shown that this unwanted side effect can be remedied by inhibiting cholesterol synthesis. BAS quite consistently increases triglycerides, while decreased triglycerides are less consistently observed upon OCA treatment. This may find its origin in the various population cohorts that were included in some of the studies, but may also be related to an incomplete understanding of the relation between plasma triglycerides and bile acid synthesis [66,67]. Early studies with other non-bile acid FXR agonists in humans, however, suggest that reductions in bile acid synthesis can also come without changes in plasma lipids, but these observations are awaiting confirmation in larger studies. This places questions about the general assumptions on the effects of FXR agonism when these are compared to those with BAS. It is in this regard, though studies in mice have suggested that FXR agonism lowers plasma lipid levels [28,29,152], this effect may be related to murine-specific changes in the physicochemical properties of the bile acid pool upon FXR stimulation [46]. Moreover, studies using FXR agonists in hamsters, that do not synthesize the mouse-/rat-specific MCAs and do express CETP, have yielded conflicting results, with decreased LDL-C being observed upon FXR inhibition [153], but also upon FXR stimulation [154,155], whereas others reported little effect of FXR stimulation [137]. Since the plasma lipid profile is clearly not improved following pharmacological FXR activation in humans, it underlines that translation of studies in animals to the human situation is challenging.

It is noteworthy to conclude that plasma lipid parameters are regularly incompletely reported and sometimes not even provided at all in the studies that we addressed for this review, which makes it difficult to evaluate how interventions in bile acid metabolism affect plasma lipids and, thereby, the risk of ASCVD in the long term. Bile acid sequestrants and FXR agonists are a positive exception in this respect, but even in these studies, plasma lipids are not always sufficiently reported. Of note, the effects of the novel bile acid modulating drugs on plasma lipid and lipoprotein metabolism certainly warrant attention because these compounds are increasingly considered for use in, for example, obese patients with T2D and NAFLD. These patients are already at high risk of developing ASCVD, an important co-morbidity because they are often characterized by increased plasma triglycerides, which is currently considered to be a causal risk factor for ASCVD [157]. Future dedicated studies into the effects of drugs that modulate bile acid metabolism or signaling pathways on especially triglyceride-rich lipoproteins in cardiometabolic patients would, in our opinion, be warranted.

## Figures and Tables

**Figure 1 jcm-11-00004-f001:**
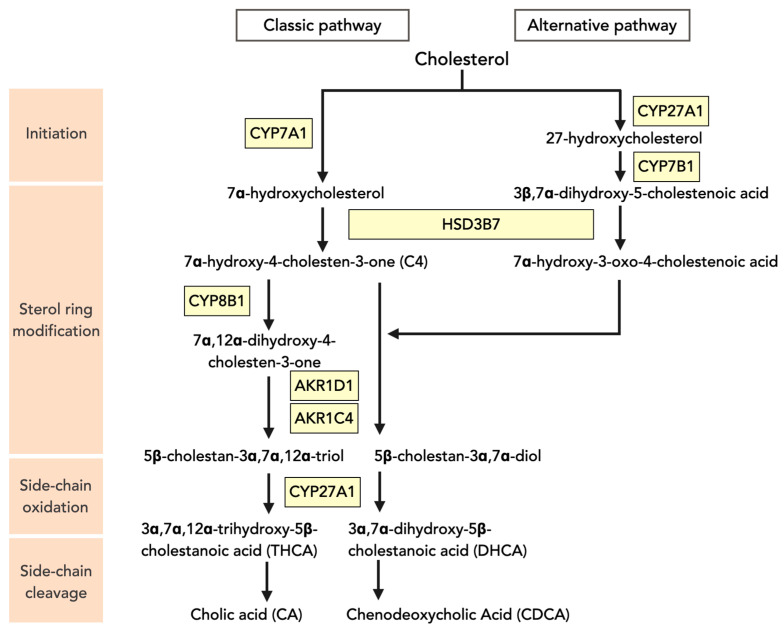
Bile acid synthetic pathways. The conversion of cholesterol into bile acids occurs through two different pathways, i.e., the classical (or neutral) pathway and the alternative (or acidic) pathway, which is quantitatively less important. The classical pathway is initiated by cholesterol 7α-hydroxylase (encoded by *CYP7A1*) in the endoplasmic reticulum of hepatocytes, and the alternative pathway is initiated by mitochondrial sterol 27-hydroxylase (encoded by *CYP27A1*). In the liver, 3β-hydroxysteroid dehydrogenase (3βHSD, HSD3B7) converts 7α-hydroxycholesterol to 7α-hydroxy-4-cholesten-3-one (C4), which is used as a surrogate marker of bile acid synthesis. Sterol 12α-hydroxylase (encoded by *CYP8B1*) introduces a hydroxyl group at the steroid nucleus (at C-12), ultimately leading to the synthesis of cholic acid (CA). Without 12α-hydroxylation, chenodeoxycholic acid (CDCA) is generated. Aldos-keto reductase 1D1 (AKR1D1) and AKR1C4 catalyse isomerization and saturation of the steroid ring. CYP27A1 then catalyses steroid side-chain oxidation. CA and CDCA represent the two main end-products of the primary bile acid synthesis pathways in the human liver and are conjugated with glycine or taurine before secretion into bile. In the intestine, these primary bile acids are subjected to activities of bacterial bile salt hydrolases and dehydroxylases, generating the various secondary bile acid species.

**Figure 2 jcm-11-00004-f002:**
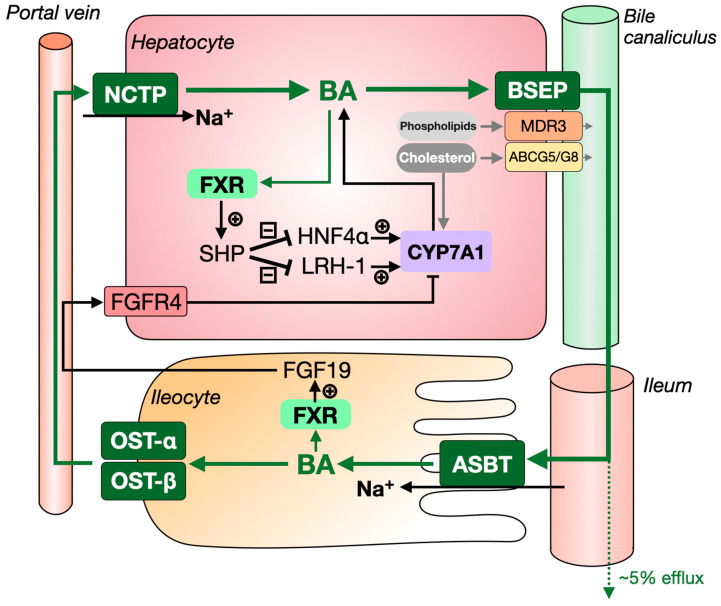
Main transporters of bile acids in the enterohepatic circulation. Once bile acids are synthesized in the liver, they are excreted to the bile canaliculus through the canalicular bile salt export pump, BSEP, and stored in the gall bladder. Phospholipids and cholesterol are transported into the bile by multidrug resistance protein 3 (MDR3) and ATP binding cassette subfamily G member 5/8 (ABCG5/G8), respectively. Bile acids are released into the intestinal lumen upon consumption of a meal to facilitate the digestion and absorption of lipids. In the terminal ileum, approximately 95% of bile acids are reabsorbed, mainly by the apical sodium-dependent bile salt transporter, ASBT. Bile acids are subsequently secreted into the portal circulation by organic solute transporter-α/β, OST-α/β and recycled to the liver, where they are mainly taken up by sodium taurocholate cotransporting polypeptide, NTCP. Bile acids cycle between the liver and intestine ~10 times/day. Each cycle, ~5% of bile acids escapes from reabsorption and is lost in the faeces. Bile acid pool size is maintained by signaling via the farnesoid X receptor (FXR), which induces transcription of its target gene small heterodimer partner (*SHP*). SHP, in turn, inhibits liver receptor homolog 1 (LRH-1) and hepatocyte nuclear factor 4 alpha (HNF4α) to reduce the expression of cytochrome P450 family 7 subfamily A member 1 (*CYP7A1*), encoding the first and rate-controlling enzyme in the major bile acid synthesis pathway. In addition, activation of FXR in the intestine induces expression of fibroblast growth factor 19 (FGF19), which is secreted into the bloodstream and activates the fibroblast growth factor receptor 4 (FGFR4) and its co-receptor β-klotho on hepatocytes, leading to inhibition of *CYP7A1* expression via incompletely elucidated mechanisms.

**Figure 3 jcm-11-00004-f003:**
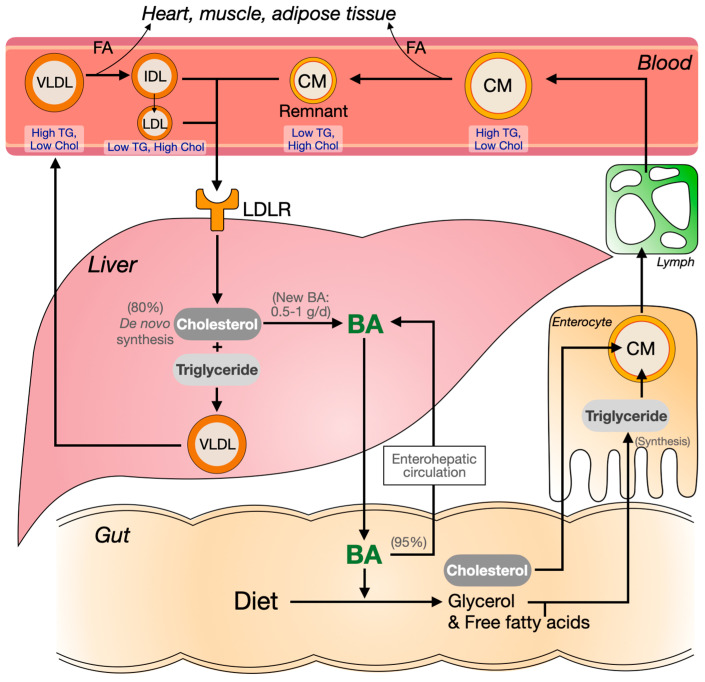
Lipid and lipoprotein metabolism. This figure gives a simplified impression of the metabolism of apoB-containing lipoproteins, which can be characterized by exogenous and endogenous pathways. Exogenous pathway: Dietary cholesterol and fatty acids (FA) are, with the help of bile acids, made available for uptake by epithelial cells of the small intestine. Here, these lipids are packaged in chylomicrons (CM), which are subsequently secreted into the lymph from where they reach the blood circulation via the thoracic duct. In the vasculature of peripheral tissues such as the heart, skeletal muscle and adipose tissue, lipoprotein lipase (LPL) catalyzes the lipolysis of triglycerides in CM, which allows for the uptake of FA by parenchymal cells. The resulting smaller so-called CM remnant particles are subsequently taken up by lipoprotein receptors such as LDLR (low-density lipoprotein receptor). Endogenous pathway: Dependent on the metabolic conditions, the liver can synthesize very-low-density lipoproteins (VLDL) to maintain overall lipid and energy homeostasis. After secretion into the systemic blood circulation, VLDL undergo the same metabolization as described for CM, which, in this case, renders smaller intermediate-density lipoproteins (IDL) and low-density lipoproteins (LDL) that are primarily taken up by the liver.

**Table 1 jcm-11-00004-t001:** Overview of BAS effects on lipid metabolism.

PMID	Therapy	Type of Patients	Number of Patient *	Year	LDL	Total Cholesterol	Non-HDL	HDL	Triglyceride	ApoB
20047620	Colestilan	T2D	86	2010	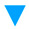			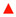	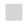	
27508319	Cholestyramine	FH and CAD	12	2016	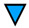	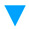	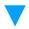	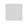	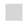	
19789153	Colesevelam	T2D	56	2009	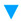	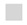	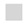	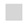	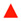	
19879596	Colesevelam	Heterozygous FH	63	2010	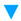	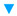	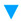	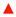	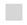	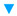
23152373	Colesevelam	Prediabetes andprimary hyperlipidaemia	103	2012	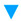	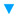	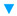	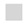	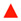	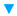
24356792	Colesevelam	T2D	176	2013	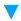	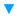	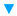		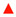	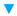
	BAS + Combination **									
15639697	Cholestyramine + Rosuvastatin	Severe hypercholesterolemia ***	76	2004	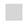	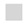		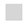	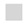	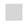
11403509	Colesevelam + Lovastatin	Primary hypercholesterolemia	27	2000	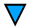	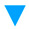		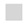	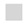	
11286949	Colesevelam + Simvastatin	Primary hypercholesterolemia	34	2001	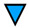	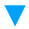		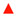	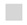	
23170931	Colesevelam + Rosuvastatin	Hypercholesterolemia andImpaired Fasting Glucose	20	2013	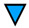	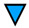	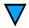	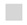	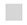	
17360295	Colesevelam + Ezetimibe	T2D with statin intolerance	18	2007	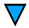	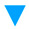	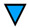	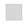	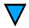	
20435231	Colesevelam + Ezetimibe + Statin	Familial Hypercholesterolemia	44	2010	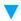	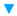		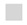	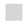	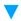
21856592	Colesevelam + Metformin	T2D	355	2011	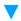	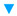	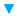	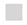	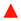	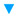
22836068	Colesevelam + Metformin	T2D	145	2012	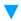	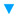	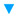		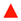	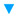
25054436	Colesevelam + pioglitazone	T2D	280	2014	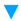	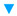	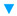	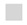	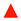	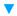
	Decrease:	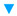	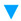	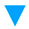	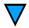		
					<10%	10–20%	20–30%	30–40%		
	Increase:	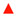	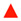	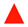	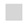		
					<10%	10–20%	>20%	No significance		

* In this table, for the BAS single-use trials, the number of subjects refers to the group that received BAS (excluding the placebo group). For combined therapy, the number of subjects refers to the group that received a high dose of BAS and other therapy (excluding the control group). ** For the combination studies, only the lipid data from the group receiving the high dose was selected. *** LDL-C level, 190–400 mg/dL.

**Table 2 jcm-11-00004-t002:** Clinical interventions on lipid metabolism by regulating bile acid metabolism.

Drugs	Mechanism of Action on Bile Acid Metabolism	Examples	Effects of Lipid Metabolism	Target Group	References
Bile acid sequestrants (BAS)	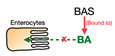	Cholestyramine, Colesevelam, Colestilan	Total cholesterol ↓ LDL-C ↓ Triglyceride ↑ HDL-C variable *	Patients with dyslipidemia, T2D, and disturbed glucose metabolism	[61,62,63,64,65]
ASBT inhibitors	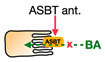	Elobixibat, Linerixibat (GSK2330672), Odevixibat (A4250) **	Total cholesterol ↓ LDL-C ↓ Triglyceride ↑ HDL-C = ***	Patients with chronic constipation, pruritus in primary biliary cholangitis, and T2D	[98,99,101,102]
FXR agonist (Steroidal)	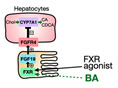	Obeticholic acid	Total cholesterol ↑ LDL-C ↑ Triglyceride ↓/= HDL-C ↓	Patients with NAFLD, NASH, T2D, and bile acid diarrhea	[126,127,128,129,130,156]
FXR agonists (Non-steroidal)		PX-102, Cilofexor, TERN-101, MET-409, GW4064 **	LDL-C =/↑ **** Triglyceride =	Healthy volunteers Patients with NASH	[132,134,139,140]
FXR agonist (Steroidal and non-steroidal)		EDP-305	LDL-C ↑ Triglyceride = HDL-C ↓	Patients with fibrotic NASH	[141]

* Increased levels of HDL-C were observed in the studies with colesevelam and colestilan while decreased levels were observed in cholestyramine. ** Plasma lipids were not measured in odevixibat/GW4064 studies. *** Equal sign indicates that no significant difference was found in the clinical studies. **** Elevated level of LDL was only found in the 80 mg/d group of MET-409 study.

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
