# Peer review of "Modulation of Bile Acid Metabolism to Improve Plasma Lipid and Lipoprotein Profiles"

_jcm, 2021, doi:10.3390/jcm11010004_

Round 1

Reviewer 1 Report

This revised manuscript has addressed the major concerns with the original manuscript and is much improved. There are, however, some points the authors may wish to consider. For example, 

  1. MDR3 and ABCG5/G8 are brought up and their roles in PC and cholesterol biliary secretion are described. However, their role to form mixed micelle with bile acids in the bile is not mentioned. The formation of mixed micelle is an important aspect should be indicated. Otherwise, MDR3 and ABCG5/G8 would appear to be brought in for no reason as separate entities.
  2. Similarly, the role of Bsep in liver steatosis is relevant to the topic but not discussed.
  3. Perhaps the role of HDL-C needs to be considered at greater length, given its apparent significance for the problem of ASCVD. The reviewers are focused on getting LDL-C down, but there isn’t an adequate mechanistic explanation given for this entire other half of the story which is summarized in one sentence (penultimate paragraph of section 1.4)

There are number of typos and inaccuracies

  • Figure 1 where the legend talks about AKR1C1 but the diagram shows AKR1C4. Is this correct?
  • Line 210: Figure 3 is described as “above” when in fact is below.
  • Line 281: “taken” should be “taking”.
  • Line 309: Noteworthily is not a word….?
  • Line 368: “is” should be “it”.
  • Line 406: reverse order of “reduction LDL-C” change to “LDL-C reduction”
  • Line 411: delete “that was”.
  • Line 643 “levels” not “levers”

Reviewer 2 Report

The introduction provides sufficient background and includes all relevant references. The methods are adequately described. The conclusions are supported by the results.

Interesting and well described review

With thorough description of the mechanisms

Author Response

Thank you for your positive comments.

Reviewer 3 Report

Very interesting  review  on a relevant  topic.

I suggest  to try to  shorten  the  manuscript , eventually  introducing  figures and  tables   to synthetize  the  data presented .

Author Response

Thank you for your comments. Addressing the concerns and suggestions in a previous review round has resulted in a comprehensive lengthy review that already includes key figures and tables. We feel that this review, in its current form, will guide the reader to sections of interest.

This manuscript is a resubmission of an earlier submission. The following is a list of the peer review reports and author responses from that submission.

Round 1

Reviewer 1 Report

Overall Assessment and General Comments

The goal of this review is to draw attention to the relationship between bile acid metabolism and plasma lipids. This is an important and timely topic that has not received much attention.  That being said, the review itself could be improved if the authors were to present a more coherent and broader mechanistic view of the interaction between bile acids and serum lipids. Rather, the message conveyed by the authors repeatedly was that the two fields are two solitudes, and that the two fields have not been collecting the necessary cross-cutting data for a more integrated view. This may be true, and is a message worth to convey. However, the authors themselves chose to limit this review primarily to human drug trials (although the authors do reference some experimental mouse work).  This is unfortunate as one wonders if some really relevant and useful mouse data that have not considered could have been helpful to make sense of, or at least speculate on some of the data from the human drug trials. Presentation of a considered mechanistic view by the authors would be a great service to the combined research fields. It could highlight areas that are experimentally solid versus areas that are in need of more research. 

There are a number of statements that are inaccurate or not properly backed up with references in this review. For instance, it is said in the Introduction, “These observations illustrate a tight connection between lipid metabolism and bile acid metabolism” referring the observations of reduction of the fecal bile acid in ASCVD patients, as if it were not for these observations, one cannot see that lipid metabolism and bile acid metabolism are closely linked. It is well known that bile acids are closely related to lipid metabolism by aiding lipid nutrient digestion and absorption, biliary clearance of lipids from the liver, etc. 

In this context the review is unbalanced in omitting the physiological role of bile acids for lipid nutrient absorption, which may, or may not affect the dynamics of atherosclerosis. The relevance of lipid absorption should be discussed more extensively. Also, the aspect of bile acids in aiding biliary secretion of lipids, and how that imbalance could change the distribution of lipids and influence the dynamics leading to atherosclerosis. That should be discussed in more detail. The extensive studies on MDR3 and ABCG5/G8 and their role in biliary secretion are not mentioned at all! A review discussing bile acid and lipid metabolism should at least acknowledge these important studies.

Specific comments and queries

  1. When the relevance of BSEP in lipid metabolism is discussed in 2.1 , the important work of Wang et al need to be cited and discussed (Wang PMID: 11172067). They have shown that the knockout of Bsep in mice increased the biliary secretion of both phospholipids and cholesterol in the KO mice. Also, the observations that BsepKO mice are more resistant to stenosis (Fuchs PMID: 32141703 and Okushin PMID: 32785220) need to be discussed.
  2. The review provides a plausible explanation for why BAS and ASBT inhibitors work as they do. However, it gets muddied when FXR agonists are discussed. There are inaccurate statements and contradictory data, some it appears based on the author’s own confusion. For example, in section 3.1 it states “OCA is a semisynthetic derivative of human bile acid chenodeoxycholic acid, which could serve as a potent FXR agonist. The drug is so far only studied to treat NAFLD, NASH, and NASH-related fibrosis:” is a highly inaccurate statement.  OCA was tested extensively in PSC, and approved for clinical use long before the more recent work on NASH. Also, in this section the authors imply OCA increasing total cholesterol is surprising. WHY? It fits with the mechanism exemplified by the bile acid sequestrates, where more bile acid = less serum LDL-C. The author may have developed this expectation only because of reference 92, the liver-specific FXR KO, even though the results of the whole body FXR KO suggest the opposite. It seems that in this case a poorly reasoned expectation leads to an artificial case of being surprised?  
  3. Specific queries
    • Page 5, paragraph 1, talks about one experiment where N has two values (reference 19).
    • Page 6 paragraph 5: blaming unclear mechanisms for contradictory data seems like they’re going too far in interpreting poor data. The data itself may be to blame.
    • Page 13, the last paragraph, “TG” - an abbreviation came out of nowhere.
    • Last paragraph second last sentence : says ASBT agonists should be antagonists
    • The review should be checked carefully for typos and inaccurate statements

Reviewer 2 Report

The review article addresses an important look at the relationship between bile acid (BA) homeostasis and plasma lipid profiles. The main target was to summarize the effect of different therapeutic approaches modulating BA homeostasis on the changes in plasma concentrations of cholesterol and triglycerides. This aim was fulfilled by the authors in terms of summarizing current studies reporting such data from human studies. Despite the very high quality of the manuscript, I have some comments and suggestions:

  1. I missed the intro where the description of BA biochemistry will be described in more detail. It is very important for understanding mechanisms and to justify the effects of agents used. Especially I am missing brief summary of the regulatory functions of FXR and TGR5 because these drugs highly interfere with insulin sensitivity, triglyceride liver synthesis and lipolysis, and cholesterol turnover.
  2. The final paragraph on page 3 duplicates the final paragraph on page 2. It would be useful to merge them and extend with regulatory part - synthesis, transporters.
  3. Other receptors for BA such as PXR, S1P2, or VDR are not mentioned.
  4. As a pharmacologist, I am quite oriented with the mechanisms of mentioned agents, but for the majority of discussed agents, these mechanisms are not clearly stated, and the information is focused on LDL/HDL-C and TAG concentrations in human studies. I like the chapter about BAS - where the data are well organized starting with paragraphs on the mechanism of action. Adding few sentences as an introduction to statins, ezetimibe, Asbt, FXR or TGR5 may greatly help the readers to understand the consequences on lipid metabolism and BA biochemistry. Also, be please more specific: e.g. TGR5 agonist paragraph. There is a sentence that"TGR5 affects bile acid, and glucose metabolism, energy expenditure, and inflammation", but particular data (way, how they are changed) are not mentioned.
  5. The relationship between bile acid homeostasis, lipid metabolism, and microbiota deserves at least a brief paragraph about mutual relationship and role in obesity.
  6. Although there are limitations in the extrapolation of rodent data to humans given to muricholic acid in BA metabolomics, and dominant HDL contrasting to LDL in humans, it may be interesting to briefly compare clinical data with the results of preclinical studies to better elucidate mechanisms.